# Zero fruits/vegetables consumption and associated factors among Children aged 6–23 months in Ethiopia: Mixed effect logistic regression analysis

**Birhan Ewunu Semagn**[1¤]*, **Abdulai Abubakari**[2]

**1** Department of Public Health, Asrat Weldeyes Health Science Campus, Debre Berhan University, Debre Berhan, Ethiopia, **2** Department of Global and International Health, School of Public Health, University for Development Studies, Tamale, Ghana

¤ Current address: Department of Social and Behavioral Change, School of Public Health, University for Development Studies, Tamale, Ghana
* ewunubirhan@gmail.com

**Data Availability Statement:** The data underlying the results presented in the study are available from Demographic and Health Survey website https://dhsprogram.com/data/.

## Abstract

### Background

The first two years of life is a vital period for promoting optimal growth, development and health. The lifelong nutritional habit and overall health of children is influenced by their early age feeding practice. Ethiopia is among the top five countries in Sub-Saharan Africa with the highest burden of zero fruits/vegetables consumption. This study aims to access factors associated with zero fruits/vegetables consumption among children aged 6–23 months in Ethiopia.

### Methods

The study analyzed Ethiopian Mini Demographic and Health Survey 2019 dataset with a total weighted sample of 1459 young children aged between 6–23 months and who were living with their mothers. Data cleaning, coding and labeling were done using STATA version 14 software. Multilevel mixed effect logistic regression model was employed to identify associated factors.

### Results

Exactly 69.3% of children aged 6–23 months in Ethiopia had zero fruits/vegetables consumption. In the multivariable multilevel binary logistic regression analysis a child from household with middle (AOR = 0.55, 95% CI: 0.35, 0.86) and rich (AOR = 0.37, 95% CI: 0.23, 0.60) wealth index, mothers who aged between 25–34 years old (AOR = 0.44; 95% CI = 0.29–0.69), mothers who were married/living with partner (AOR = 3.21; 95%CI: 1.58–6.52), children of mothers who follow Islamic religion (AOR = 0.34, 95% CI: 0.19, 0.61), mothers who had more than four ANC visits during their most recent pregnancy (AOR = 0.57; 95%CI: 0.39–0.83), children in age group of 12–18 month(AOR = 0.41, 95% CI:

**Funding:** The authors received no specific funding for this work.

**Competing interests:** The authors have declared that no competing interests exist.

**Abbreviations:** ANC, Antenatal Care; AOR, Adjusted Odds Ratio; CI, Confidence Interval; COR, Crude Odds Ratio; DHS, Demographic and Health Survey; EDHS, Ethiopian Demographic and Health Survey; EMDHS, Ethiopian Mini Demographic and Health Survey; UNICEF, United Nations Children's Fund; WHO, World Health Organization.

0.28, 0.59), and 19–23 months (AOR = 0.26, 95% CI: 0.17, 0.40), health facility delivery (AOR = 1.52, 95% CI; 1.00–2.30), and small peripheral regions (AOR = 4.40, 95% CI; 1.39–13.97) were found to be significant factors associated with children's zero fruits /vegetables consumption. The Interclass correlation coefficient (ICC) value in the null model was 0.34, which indicates that 34% of the variation in children's zero fruits /vegetables consumption was attributed to the variation between clusters.

## Conclusion

This study found that zero fruits/vegetables consumption among children aged 6–23 months in Ethiopia is high. Therefore, efforts should be made by stakeholders who are concerned about optimal diet and health of children to improve fruits/vegetables consumption of children particularly those from poor households, young mothers (15–24), and peripheral regions of Ethiopia. This could be done during ANC follow up visits and during nutrition counseling.

## Background

The first two years of life which is characterized by rapid growth rate, higher energy and nutrient requirement is a vital period for promoting optimal growth, development and health during early life [1]. Feeding practice during early life contributes to lifelong nutritional habit, and overall health [2, 3]. Apart from short-term consequences of micronutrient deficiencies and diseases, inadequate consumption of fruits and vegetables also increases the risk of non-communicable diseases like cancer, cardio-vascular disease, obesity, and metabolic diseases like diabetes mellitus [4, 5]. Moreover, scientific evidences available shows that consumption of fruits and vegetables prevent cataract formation, coronary heart disease, stroke, chronic obstructive pulmonary disease, diverticulosis, cancer, depression and improve cognitive and mental health [6–10].

Even though there is no universal recommendation for the optimal number of servings of vegetables and fruits per day for infants and young children over six months of age, the World Health Organization (WHO) and the United Nations Children's Fund (UNICEF) consider consumption of zero vegetables or fruits on the previous day as unhealthy child feeding practice [11]. This indicator was added to the existing Infant and young child-feeding practice indicators during the UN International Year of Fruits and Vegetables in 2021 [11].

Generally, worldwide fruits and vegetables consumption is below the recommended level [12]. The burden of poor diets and the low fruits and vegetables consumption is high in low and middle-income countries [13–15]. According to previous studies conducted at a sub-continent level, 47% of children in Sub-Saharan Africa had zero fruits/vegetables consumption [16].

Ethiopia is among the top five countries in Sub-Saharan Africa which has highest burden of zero fruits/vegetables consumption (69%) among children [16]. Previous studies which were conducted to assess dietary diversity among children in Ethiopia investigated the consumption of fruits and vegetables rich in Vitamin A as a single food group, and other fruits and vegetables as another food group and, among other food groups as part of the overall diet. Most of these studies demonstrate low consumption of fruits and vegetables among children age 6–23 months [17–22]. For example a study conducted in southern Ethiopia reported 7.2% and 39.1% respectively of children who consumed Vitamin A rich, and other fruits and vegetables

the day prior to the survey [18]. Another study in southern Ethiopia found 21.5% and 11.8%) consumption of vitamin A-rich fruits and vegetables, and other fruits and vegetables [17]. Furthermore, a nationwide study which used Ethiopian Demographic and Health Survey 2016 (EDHS) reported 25.1%-32.8% and 28.1%-31.6% of Vitamin-A rich fruits and vegetables, and other fruits and vegetables consumptions respectively [23].

Previous studies identified child's age, mothers' age, media exposure, maternal working status, household wealth, ecology, distance to the health facility, number of births a mother have in the last five years, and residence as factors significantly associated with children's zero fruits/vegetables consumption [16, 24].

Though the Ethiopian Mini Demographic and Health Survey (EMDHS) report capture children's consumption of vitamin-A rich foods and other fruits/vegetables consumption, it did not show the burden of zero fruits/vegetables consumption and associated factors [25].

In addition, while zero fruits/vegetables consumption by children is one of the indicators for assessing infant and young child feeding practices by WHO and the UNICEF, there is dearth of knowledge on the burden of zero fruits/vegetables consumption by children and associated factors in Ethiopia. Therefore, the present study aims to fill this research gap using the most recent nationally representative data. Besides, finding may serve as baseline to further develop an interventional study that will contribute for the big evidence gap on how to improve fruits and vegetables consumption of children [4, 13, 26].

## Methods

### Study design, data source, and setting

This study is an analytical cross-sectional study that used data from the most recent national representative EMDHS 2019. The 2019 EMDHS was the second EMDHS and the fifth EDHS implemented by the Ethiopian Public Health Institute (EPHI), in partnership with the Central Statistical Agency (CSA) and the Federal Ministry of Health (FMOH). Data collection lasted from March 21, 2019, to June 28, 2019. For this study the data were obtained from the Demographic and Health survey (DHS) website (https://dhsprogram.com/) after submitting a request justifying the aim of the study. We used the Kids Record (KR) file of EMDHS data set which contains information related to pregnancy, postnatal care, immunization, health and nutrition data of mother child pair. Ethiopia is one of the low-income Sub-Saharan countries in East Africa. At the time of the data collection Ethiopia had nine geographical regions and two administrative cities [25].

### Population and sampling procedure

This study is based on a weighted sample of 1459 young children aged between 6–23 months who were living with their mother (KR file), and born in the 2 years preceding the survey (Fig 1). The 2019 EMDHS used two stage stratified sampling technique. In the first stage, a total of 305 Enumeration Areas (EAs); 93 in urban areas and 212 in rural areas: were selected with probability proportional to EA size. In the second stage of selection, a fixed number of 30 households per cluster were selected with an equal probability using systematic sampling technique. All women aged 15–49 who were either permanent residents of the selected households or visitors who slept in the household the night before the survey were eligible to be interviewed. Finally, 8,885 women aged between 15–49 years old were interviewed. Further information related to the population, study area, data collection, sampling procedure, and questionnaires used in the survey were detailed in the 2019 EMDHS Report [25].

Household interviewed in 2019 EMDHS= 8,663

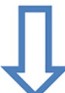

Weighted number of reproductive aged women interviewed in 2019 EMDHS = 8,885

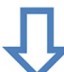

Weighted number youngest children age 6-23 months living with their mother in 2019 EMDHS =1463

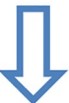

Weighted number of youngest children age 6-23 months living with their mother included in this study=1459*

**Fig 1. Illustrates the sampling procedure, and the final sample size considered in the study to assess the proportions of zero fruits/vegetables consumption, and associated factors among children aged 6–23 months in Ethiopia using 2019 EMDHS dataset.** * 4 of the respondents responded don't know for the fruit/vegetable consumption of their children.

### Study variables

**Outcome variable.**   Zero fruits/vegetables consumption.

This is the proportion of children who did not consume any vegetable or fruit during the previous day. This indicator is based on consumption of food groups (vitamin A-rich fruits and vegetables, and other fruits and vegetables). These were part of the 24-hour dietary recall used to estimate Minimum Dietary Diversity (MDD) in the DHS data set. Children were counted if there was no consumption of either food groups [11].

**Independent variables.**   Variables were selected by consulting literature [16, 23, 27–30] and based on their availability in DHS dataset. Individual level variables were, maternal age, maternal marital status, maternal education, religion, sex of a child, child age, birth order, Antenatal Care (ANC) by skilled provider, having four or more ANC, ANC

initiation in the first trimster of pregnancy, place of delivery, baby post natal check within two month, and having a son or daughter died. Household level variables like household wealth index (the EMDHS datasets had a variable labeled as wealth index and coded as "poorest", "poorer", "middle", "richer", and "richest". However, in the present study wealth index was recorded into three categories, "poor" (includes the poorest and the poorer categories), "middle", and "rich" (includes the richer and the richest categories), media access (a composite variable obtained by combining whether there was a radio and /or TV in the respondent's house; with a value of"0" if a woman didn't have either TV or Radio in her household and "1" if a woman has access to either of the media), household size, Community level variables such as residence and region were (grouped in to three categories: metropolitan Harar and Drie-Dawa, large central regions Amhara, Oromia, South Nations and nationalities and Tigray, and Small peripheral regions Afar, Benishangul, Gambella, and Somalia).

## Data management and analysis

After accessing the data from DHS website data extraction, recoding, labeling, cross-tabulations and analysis were done using STATA Version 14. Prior to conducting any statistical analysis, the data was weighted using sampling weight (v005/1000000), primary sampling unit (v021), and strata (v022) to keep the representativeness of the survey and to get more reliable estimates. Frequencies and percentages were used for descriptive statistics. For the inferential statistics we used multi-level logistic regression analysis because the hierarchical structure of the DHS data violates the assumption of independent observation and equal variance [25].

Assessment of community level clustering, and model comparison were done. The Interclass Correlation Coefficient (ICC) and Median Odds Ratio (MOR) were checked to assess whether there was clustering or not. Four separate models including the null model, model I (individual-level variables), model II (household and community-level variable), and Model III (models that include both individual, household and community level variables) were fitted. Model comparison were done using deviance test. The best fitted model with the lowest deviance, was chosen Table 2.

Finally, both bi-variable and multivariable multi-level logistic regression analysis were done using the best-fitted model (model III which included both individual, household and community level variables). Variables with a p-value ≤0.25 at the bi-variable analysis were considered for multivariable analysis, and variables with a p-value of ≤0.05 in the multivariable analysis were considered statistically significant. The STORBE guideline was followed in writing the manuscript [31].

## Ethical consideration

The present study used publicly available secondary data from DHS website. Permission was granted to access and use the data after submitting an online request that justify the objectives of the study. Ethical clearance was sought by the institutions that funded, commissioned, and managed the survey, and no further ethical clearance was needed. The Institutional Review Board approved procedures for DHS public-use of datasets that does not allow respondents, or households to be identified. Any personal identifiers like names, house numbers, and phone numbers were not included in the dataset. Furthermore, as the study was based on secondary data analysis, gaining participants consent was not applicable.

## Results

### Socio-demographic characteristics, and proportion of zero fruits /vegetables consumption

**Socio-demographic characteristics.** This study was based on a weighted sample of 1459 youngest children age 6–23 months who were living with their mother. Most of the children were from rural areas (71.7%), from large central regions (86.8%), had mothers aged 25–34 years (49.5%), had their mothers married/living with partner (95.3%) and, had mothers with no formal education (44.4%). Most of the children also had mothers with no media access (64.4%), who belong to orthodox religion (36.9%), who received ANC from skilled provider (75.6%), who attended four or more ANC (54.6%), and who did not attend ANC in their first trimester of pregnancy (70.9%). Majority of the study children were from households with poor wealth index (41.4%), household size of less than or equal to five (55.0%), and headed by males (86.1%). Moreover, the higher proportion of the study children were males (52.3%), within the age group of 12–18 months (43.9%), 2–4 birth order (47.3%), born in a health facility (55.1%), and with no post-natal check within two months (86.4%) of their birth (Table 1).

**Proportion of zero fruits /vegetables consumption.** The proportion of zero fruits/vegetables consumption was 69.3% [95% CI = 64.56, 73.71]. The proportion of zero fruits/vegetables consumption is higher among children of mothers who reside in rural areas (50.9%), large central region (59.1%), had no formal education (33.5%), had no media access (48.0%), who did not attend four or more ANC (40.5%), and who did not had first trimester ANC visit (50.8%). The proportion of zero fruits/vegetables consumption was also higher among children from poor household's wealth index (33.04%), with household size of less than or equal to five (36.9%), and of household headed by males (58.9%). Higher proportion of zero fruits/vegetables consumption was observed among children aged 12–18 months (30.4%) and with no post-natal care within two months of birth (60.8%) (Table 1).

### Random effect (community-level clustering) and model comparison

The existence of community level clustering was assessed using the random effect model. The result of random effect model implies the presence of significant clustering given that the ICC value in the null model was 0.34, which indicates that about 34% of the total variation in zero fruits/vegetables consumption was attributable to the variation between clusters, moreover the value of MOR (95% CI) was 3.45(2.58–4.32) which shows that zero fruits/vegetables consumption among children is significantly different between clusters. This means that, if we randomly choose children between the age of 6 and 23 months from various clusters, those from the cluster with highest zero fruits/vegetables consumption had 3.45 times the odds of doing so compared to those from the cluster with the lowest zero fruits/vegetables consumption.

Regarding model fitness, the final model (Model III) with lowest deviance was the best model for forecasting the association of independent variables with zero fruits/vegetables consumption (Table 2).

### Individual, household and community level factors associated with zero fruits /vegetables consumption among children aged 6–23 months

The variables with p-value< 0.25 in the bi-variable multi-level logistic regression analysis were residence, region, wealth-index, household family size, mother's age, mother's marital status, mother's educational level, media access, religion, sex of a child, child age, birth order, ANC from skilled provider, attended 4+ ANC visits, attended ANC in the first trimester of

**Table 1. Socio-demographic characteristics of caregivers/mothers and their children aged 6–23 months by children's zero fruits /vegetables consumption.**

Zero 24 hours fruits/vegetables consumption

| Variables | No | | Yes | | Total | | Weighted Frequency |
|---|---|---|---|---|---|---|---|
| | % | CI | % | CI | % | CI | |
| **Type of place of residence** | | | | | | | |
| Urban | 9.8 | [6.99,13.56] | 18.47 | [13.94,24.06] | 28.27 | [23.54,33.53] | 412 |
| Rural | 20.88 | [17.55,24.65] | 50.86 | [45.83,55.87] | 71.73 | [66.47,76.46] | 1,047 |
| Total | 30.67 | [26.29,35.44] | 69.33 | [64.56,73.71] | 100 | | 1,459 |
| **Geopolitical features of regions** | | | | | | | |
| Metropolitans | 1.94 | [1.47,2.56] | 2.2 | [1.69,2.87] | 4.15 | [3.38,5.07] | 60 |
| Small peripheral regions | 1 | [0.78,1.28] | 8.07 | [6.30,10.29] | 9.07 | [7.28,11.26] | 132 |
| Large central regions | 27.73 | [23.44,32.47] | 59.05 | [54.02,63.90] | 86.78 | [84.38,88.86] | 1,266 |
| Total | 30.67 | [26.29,35.44] | 69.33 | [64.56,73.71] | 100 | | 1,459 |
| **Wealth-index** | | | | | | | |
| Poor | 8.34 | [6.36,10.88] | 33.04 | [27.65,38.91] | 41.38 | [35.22,47.82] | 604 |
| Middle | 6.84 | [4.91,9.47] | 12.03 | [9.22,15.56] | 18.88 | [15.36,22.97] | 275 |
| Rich | 15.49 | [11.95,19.83] | 24.26 | [19.30,30.02] | 39.74 | [34.04,45.75] | 580 |
| Total | 30.67 | [26.29,35.44] | 69.33 | [64.56,73.71] | 100 | | 1,459 |
| **House hold size** | | | | | | | |
| Greater than five | 12.54 | [9.73,16.02] | 32.44 | [28.54,36.61] | 44.98 | [40.61,49.44] | 656 |
| Less than or equal to five | 18.13 | [14.74,22.11] | 36.88 | [32.48,41.52] | 55.02 | [50.56,59.39] | 803 |
| Total | 30.67 | [26.29,35.44] | 69.33 | [64.56,73.71] | 100 | | 1,459 |
| **Sex of household head** | | | | | | | |
| Female | 3.44 | [2.34,5.03] | 10.46 | [7.76,13.96] | 13.9 | [11.17,17.18] | 203 |
| Male | 27.23 | [22.79,32.18] | 58.86 | [54.52,63.07] | 86.1 | [82.82,88.83] | 1,256 |
| Total | 30.67 | [26.29,35.44] | 69.33 | [64.56,73.71] | 100 | | 1,459 |
| **Mother's age category** | | | | | | | |
| 15–24 | 9.15 | [7.15,11.64] | 23.16 | [19.70,27.03] | 32.31 | [28.36,36.54] | 471 |
| 25–34 | 16.76 | [13.87,20.11] | 32.76 | [29.18,36.56] | 49.52 | [45.84,53.21] | 723 |
| > = 35 | 4.76 | [3.27,6.89] | 13.4 | [10.77,16.56] | 18.16 | [15.49,21.19] | 265 |
| Total | 30.67 | [26.29,35.44] | 69.33 | [64.56,73.71] | 100 | | 1,459 |
| **Marital status** | | | | | | | |
| Never in union/no longer living together | 1.81 | [0.94,3.48] | 2.87 | [1.82,4.49] | 4.68 | [3.13,6.94] | 68 |
| Married/living with partner | 28.86 | [24.37,33.80] | 66.46 | [61.98,70.66] | 95.32 | [93.06,96.87] | 1,391 |
| Total | 30.67 | [26.29,35.44] | 69.33 | [64.56,73.71] | 100 | | 1,459 |
| **Mother's education** | | | | | | | |
| No education | 10.93 | [8.22,14.39] | 33.51 | [29.37,37.92] | 44.44 | [39.54,49.44] | 648 |
| Primary | 13.59 | [10.91,16.80] | 28.16 | [24.57,32.04] | 41.74 | [37.83,45.77] | 609 |
| Secondary | 3.69 | [2.32,5.82] | 4.52 | [3.15,6.45] | 8.21 | [6.27,10.67] | 120 |
| Higher | 2.47 | [1.61,3.77] | 3.14 | [1.91,5.14] | 5.61 | [4.02,7.79] | 82 |
| Total | 30.67 | [26.29,35.44] | 69.33 | [64.56,73.71] | 100 | | 1459 |
| **Media access (radio / TV)** | | | | | | | |
| No media access | 16.32 | [13.21,19.99] | 48.03 | [43.31,52.79] | 64.35 | [59.19,69.20] | 934 |
| Has media access | 14.24 | [10.57,18.92] | 21.4 | [18.26,24.92] | 35.65 | [30.80,40.81] | 517 |
| Total | 30.56 | [26.17,35.35] | 69.44 | [64.65,73.83] | 100 | | 1,451* |
| **Religion** | | | | | | | |
| Orthodox | 9.52 | [7.00,12.84] | 27.35 | [22.31,33.03] | 36.87 | [30.87,43.31] | 538 |
| Muslim | 11.53 | [7.56,17.19] | 20.98 | [16.40,26.42] | 32.5 | [24.93,41.11] | 474 |
| Protestant | 9.41 | [6.48,13.49] | 18.79 | [13.62,25.34] | 28.2 | [20.81,36.98] | 411 |

*(Continued)*

**Table 1.** (Continued)

**Zero 24 hours fruits/vegetables consumption**

| Variables | No | | Yes | | Total | | Weighted Frequency |
|---|---|---|---|---|---|---|---|
| | % | CI | % | CI | % | CI | |
| Catholic, traditional and other | 0.21 | [0.06,0.74] | 2.22 | [0.80,6.00] | 2.43 | [0.95,6.03] | 35 |
| Total | 30.67 | [26.29,35.44] | 69.33 | [64.56,73.71] | 100 | | 1,459 |
| **Sex of a child** | | | | | | | |
| Male | 15.67 | [12.51,19.45] | 36.67 | [32.48,41.07] | 52.34 | [48.58,56.07] | 764 |
| Female | 15 | [12.27,18.22] | 32.66 | [28.59,37.00] | 47.66 | [43.93,51.42] | 695 |
| Total | 30.67 | [26.29,35.44] | 69.33 | [64.56,73.71] | 100 | | 1,459 |
| **Child age category** | | | | | | | |
| 6–11 | 7.45 | [5.52,9.99] | 25.14 | [22.47,28.02] | 32.59 | [29.50,35.84] | 476 |
| 12–18 | 13.44 | [10.99,16.34] | 30.43 | [26.06,35.18] | 43.87 | [39.52,48.31] | 640 |
| 19–23 | 9.78 | [7.36,12.88] | 13.76 | [11.21,16.78] | 23.54 | [19.87,27.65] | 343 |
| Total | 30.67 | [26.29,35.44] | 69.33 | [64.56,73.71] | 100 | | 1,459 |
| **Birth order** | | | | | | | |
| 1 | 7.93 | [6.17,10.15] | 16.27 | [12.97,20.21] | 24.2 | [20.48,28.35] | 353 |
| 2–4 | 15.64 | [12.89,18.85] | 31.64 | [27.53,36.05] | 47.28 | [43.06,51.53] | 690 |
| > = 5 | 7.1 | [5.12,9.77] | 21.42 | [18.29,24.93] | 28.52 | [24.94,32.40] | 416 |
| Total | 30.67 | [26.29,35.44] | 69.33 | [64.56,73.71] | 100 | | 1,459 |
| **ANC from skilled provider** | | | | | | | |
| No | 5.66 | [3.87,8.22] | 18.7 | [15.43,22.48] | 24.36 | [20.68,28.47] | 355 |
| Yes | 25.01 | [20.75,29.81] | 50.63 | [45.87,55.37] | 75.64 | [71.53,79.32] | 1,104 |
| Total | 30.67 | [26.29,35.44] | 69.33 | [64.56,73.71] | 100 | | 1,459 |
| **Attended 4+ ANC visits** | | | | | | | |
| No | 14.14 | [11.31,17.54] | 40.5 | [36.17,44.98] | 54.64 | [50.29,58.91] | 794 |
| Yes | 16.46 | [13.32,20.17] | 28.9 | [25.33,32.76] | 45.36 | [41.09,49.71] | 660 |
| Total | 30.6 | [26.22,35.36] | 69.4 | [64.64,73.78] | 100 | | 1,454* |
| **Attended ANC in the first trimester of pregnancy** | | | | | | | |
| No | 20.09 | [16.92,23.69] | 50.8 | [46.12,55.46] | 70.89 | [67.51,74.05] | 1,034 |
| Yes | 10.58 | [8.19,13.57] | 18.53 | [15.92,21.46] | 29.11 | [25.95,32.49] | 425 |
| Total | 30.67 | [26.29,35.44] | 69.33 | [64.56,73.71] | 100 | | 1,459 |
| **Place of delivery** | | | | | | | |
| Non-health facility | 12.01 | [9.10,15.69] | 32.89 | [27.67,38.57] | 44.9 | [38.18,51.80] | 655 |
| Health facility | 18.66 | [14.43,23.80] | 36.44 | [31.52,41.66] | 55.1 | [48.20,61.82] | 804 |
| Total | 30.67 | [26.29,35.44] | 69.33 | [64.56,73.71] | 100 | | 1,459 |
| **Baby postnatal check within 2 months** | | | | | | | |
| No | 25.62 | [21.56,30.16] | 60.77 | [55.54,65.77] | 86.39 | [82.94,89.24] | 1,257 |
| Yes | 5.13 | [3.25,8.01] | 8.47 | [6.59,10.83] | 13.61 | [10.76,17.06] | 198 |
| Total | 30.76 | [26.29,35.62] | 69.24 | [64.38,73.71] | 100 | | 1,455* |
| **Given birth to a boy or girl who was born alive but later died** | | | | | | | |
| No | 30.22 | [25.89,34.93] | 67.37 | [62.69,71.72] | 97.58 | [96.32,98.42] | 1,424 |
| Yes | 0.45 | [0.15,1.35] | 1.96 | [1.22,3.14] | 2.42 | [1.58,3.68] | 35 |
| **Total** | **30.67** | **[26.29,35.44]** | **69.33** | **[64.56,73.71]** | **100** | | **1,459** |

*not dejure resident, and don't know were considered as missing and deleted from further analysis

**Table 2. Random effect model and model fitness comparison for factors associated with children's zero fruits /vegetables consumption, Ethiopia.**

| Parameter | Null model | Model I | Model II | Model III * |
|---|---|---|---|---|
| Variance (SE) | 1.69(.34) | 2.01 (.43) | 1.05(.25) | 1.86(.42) |
| ICC | 0.34 | 0.38 | 0.24 | 0.36 |
| MOR (95%CI) | 3.45(2.58 4.32) | 3.9(2.76 4.96) | 2.65(2.05 3.26) | 3.68(2.62 4.74) |
| PCV | Reference | -0.19 | 0.38 | -0.1 |
| | | Model comparison | | |
| LL | -814.39 | -751 | -803.92 | -725.67 |
| Deviance | 1628.78 | 1502 | 1607.84 | 1451.34 |

*the best fitted model with lowest deviance

pregnancy, place of delivery, baby post-natal check within two month, and given birth to a boy or girl who was born alive but later died (Table 3).

In the multi-variable multi-level logistic regression analysis wealth-index, mother's age, mother's marital status, religion, child age, attended 4+ ANC visit, place of delivery and region were found to be significant factors associated with children's zero fruits /vegetables consumption. A child from household with middle and rich wealth index had 55% and 37% less odds of zero fruits/vegetables consumption (AOR = 0.55, 95% CI: 0.35, 0.86) and (AOR = 0.37, 95% CI: 0.23, 0.60) respectively compared to children with poor household index.

Children of mothers who aged between 25–34 years old were 56% times less likely to have zero fruits/vegetables consumption than children from mothers aged between 15–24 years old (AOR = 0.44; 95%CI = 0.29–0.69). Also, children of mothers who were married/living with partner had 3.21 (AOR = 3.21; 95%CI: 1.58–6.52) times higher odds of zero fruits/vegetables consumption compared to children of mothers/caregivers who have never been in a union/no longer living together. Moreover, children of mothers, who follow Islamic religion, had 34% lower odds of zero fruits/vegetables consumption (AOR = 0.34, 95% CI: 0.19, 0.61) compared to those of mother's who follow Orthodox religion.

Further, the study revealed that the odds of zero fruits/vegetables consumption among children aged 12–18 month, and 19–23 months were 41% (AOR = 0.41, 95% CI: 0.28, 0.59), and 26% (AOR = 0.26, 95% CI: 0.17, 0.40) respectively, lower compared to children aged 6–11 months.

Moreover, children of mothers who had more than four ANC visits during their most recent pregnancy had 57% (AOR = 0.57; 95%CI: 0.39–0.83) lower odds of zero fruits/vegetables consumption compared to their counterparts. The study also shows higher odds of zero fruits/vegetables consumption among children who were born in health facility compared to their counterparts (AOR = 1.52, 95% CI; 1.00–2.30).

Furthermore, among the community level variables region was identified as significant factor associated with zero fruits/vegetables consumption as children from small peripheral regions had 4.40 times (AOR = 4.40, 95% CI; 1.39–13.97) higher odds of zero fruits/vegetables consumption compared to children from the metropolitan regions (Table 3).

## Discussion

Zero fruits /vegetables consumption is one of the UNICEF/WHO indicators of infant and young child feeding practices. Among Ethiopian children age 6–23 months, 69.3% did not consume any vegetables or fruits a day preceding the survey. The proportion of zero fruits/vegetables consumption in our study is lower than a study that reported 77.5% and 76.3% of

**Table 3. Bi-variable, and multi-variable multilevel logistic regression analysis of individual, household and community level factors associated with children's zero fruits /vegetables consumption.**

| Variables | COR | 95%CI | AOR | 95%CI |
|---|---|---|---|---|
| **Type of place of residence** | | | | |
| Urban | 1 | | 1 | |
| Rural | 1.94*** | 1.11–3.36 | 0.96 | 0.46–1.99 |
| **Geopolitical features of regions** | | | | |
| Metropolitans | 1 | | | |
| Small peripheral regions | 7.96*** | 3.00–21.15 | 4.40* | 1.39–13.97 |
| Large central regions | 2.15*** | 1.02–4.54 | 1.00 | 0.38–2.63 |
| **Wealth-index** | | | | |
| Poor | 1 | | 1 | |
| Middle | 0.51*** | 0.34–0.77 | 0.55** | 0.35–0.86 |
| Rich | 0.31*** | 0.22–0.45 | 0.37*** | 0.23–0.60 |
| **House hold size** | | | | |
| Greater than 5 | 1 | | 1 | |
| Less than or equal to five | 0.68*** | 0.51–0.90 | 0.80 | 0.54–1.19 |
| **Sex of household head** | | | | |
| Female | 1 | | | |
| Male | 1.02 | 0.67–1.55 | | |
| **Mother's age category** | | | | |
| 15–24 | 1 | | 1 | |
| 25–34 | 0.72*** | 0.53–0.99 | 0.44*** | 0.29–0.69 |
| > = 35 | 1.16 | 0.76–1.75 | 0.56 | 0.30–1.06 |
| **Marital status** | | | | |
| Never in union/no longer living together | 1 | | 1 | |
| Married/living with partner | 2.21*** | 1.15–4.26 | 3.21** | 1.58–6.52 |
| **Mother's Education** | | | | |
| Primary | 0.74*** | 0.54–1.00 | 0.84 | 0.58–1.22 |
| Secondary | 0.38*** | 0.23–0.63 | 0.58 | 0.31–1.11 |
| Higher | 0.36*** | 0.19–0.66 | 0.69 | 0.32–1.47 |
| **Media access (radio / TV)** | | | | |
| No media access | 1 | | 1 | |
| Has media access | 0.56*** | 0.41–0.76 | 0.93 | 0.64–1.36 |
| **Religion** | | | | |
| Orthodox | 1 | | 1 | |
| Muslim | 0.90 | 0.55–1.46 | 0.34*** | 0.19–0.61 |
| Protestant | 1.03 | 0.62–1.72 | 0.74 | 0.42–1.31 |
| Catholic, traditional and other | 4.18*** | 1.04–16.77 | 2.97 | 0.71–12.46 |
| **Sex of a child** | | | | |
| Male | 1 | | 1 | |
| Female | 0.81*** | 0.62–1.06 | 0.77 | 0.57–1.04 |
| **Child age category** | | | | |
| 6–11 | 1 | | 1 | |
| 12–18 | 0.49*** | 0.35–0.68 | 0.41*** | 0.28–0.59 |
| 19–23 | 0.31*** | 0.21–0.45 | 0.26*** | 0.17–0.40 |
| **Birth order** | | | | |
| 1 | 1 | | 1 | |
| 2–4 | 1.03 | 0.73–1.45 | 1.16 | 0.74–1.81 |

(*Continued*)

**Table 3.** (Continued)

| Variables | COR | 95%CI | AOR | 95%CI |
|---|---|---|---|---|
| > = 5 | 1.72*** | 1.16–2.55 | 1.87 | 0.96–3.64 |
| **ANC from skilled provider** | | | | |
| No | 1 | | 1 | |
| Yes | 0.56*** | 0.39–0.80 | 0.93 | 0.59–1.48 |
| Total | | | | |
| **Attended 4+ ANC visits** | | | | |
| No | 1 | | 1 | |
| Yes | 0.59*** | 0.44–0.79 | 0.57** | 0.39–0.83 |
| **Attended ANC in the first trimester of pregnancy** | | | | |
| No | 1 | | 1 | |
| Yes | 0.79*** | 0.59–1.06 | 1.17 | 0.81–1.69 |
| **Place of delivery** | | | | |
| Non-health facility | 1 | | 1 | |
| Health facility | 0.79*** | 0.57–1.09 | 1.52* | 1.00–2.30 |
| **Baby postnatal check within 2 months** | | | | |
| No | 1 | | 1 | |
| Yes | 0.77*** | 0.52–1.12 | 0.77 | 0.50–1.19 |
| **Given birth to a boy or girl who was born alive but later died** | | | | |
| No | 1 | | 1 | |
| Yes | 2.07*** | 0.75–5.66 | 1.43 | 0.49–4.19 |
| | **COR*** p<0.25** | | **AOR *** p<0.001, ** p<0.01, * p<0.05** | |

school adolescent respectively in Seven African countries and five southeast Asian countries consumed less than the recommended five servings of fruits and/or vegetable a day [27, 32]. The finding of the present study was also lower than the finding of a study conducted to assess global variability of fruits and vegetables consumption, which reported that "78.0% of respondents from mainly low- and middle-income countries consumed less than the minimum recommended five daily servings of fruits and vegetables" [33]. The proportion of zero fruits/vegetables consumption in the present study is higher than a study conducted among in-school adolescents in southeast Asian countries that revealed 28% consuming fruits less than once per day and 13.8% of the participant consuming vegetables less than once per day [32]. These discrepancies might be due to the difference in the, socio-demographic characteristics, geographic, climate and feeding habit of the respondents.

In multivariable multilevel logistic regression analysis household wealth index, mother's age, mother's marital status, religion, child age, attending 4+ANC, place of delivery and region were found to be statistically significantly associated with zero fruits/vegetables consumption among children aged 6–23 months.

In the present study wealth index was significantly associated with zero fruits/vegetables consumption. A child from household with middle and rich wealth index had low probability of zero fruits/vegetables consumption compared to children from poor households. This is in line with a study conducted to assess social in equality in fruits and vegetables consumption by household and socio-demographic characteristics in Argentina and Thailand [24, 34]. This might be because of several reasons. First of all, it might be because of access to healthy foods, children from middle- and high-income households may have better access to fruits and vegetables because their families can afford to purchase them. On the other hand, families with lower income may have limited access to healthy food options due to their financial

constraints. Secondly the household's food insecurity could be the reason for high zero fruits and vegetables consumption of children from poor household wealth index. Family with lower incomes may have poor access to food, which can lead to lack of access to healthy foods, including fruits and vegetables. Furthermore, it could be due to better complementary feeding practice of women from middle and rich household wealth index [35]. Therefore, financially empowering and general socio-economic conditions of households could enhance household food security and for that matter fruits and vegetable consumption. Moreover, promoting complementary feeding for children will reduce zero fruits/vegetables consumption.

Mother's age and marital status was also found to be significant predictors of zero fruits/vegetables consumption. Children of mothers aged between 25–34 years old were less likely to have zero fruits/vegetables consumption compared to children of mother aged 15–24 years old. Also, children of mothers who were married/living with their partners had higher probability of zero fruits/vegetables consumption. This is in line with a study conducted among children of East and South African Countries that reported that being a young mother (15–24 years) is a risk factor for not meeting minimum dietary diversity for children of East Africa [36]. This might be related to their experience in feeding young children as older mothers are more likely to have more experience in practicing complementary feeding and parenting than younger mothers. It could also be that older mothers have more financial muscles and attained higher educational level, which may enhance their ability to access variety of foods for their children compare to younger ones. Also, mothers in the 25–34 age group may have had more education and exposure to nutrition education programs, which can increase their knowledge and understanding of the importance of fruits and vegetables in their children's diets. Therefore, multisectoral collaboration to address the academic, and financial needs of mothers could help reduce zero fruits/vegetables consumption among children. Furthermore, giving training on parenting and feeding of children will contribute to the reduction of zero fruits/vegetables consumption.

Frequency of ANC service utilization and place of delivery were also associated with zero fruits/vegetables consumption. Children of mothers who had more than 4 ANC visit were less likely to have zero fruits/vegetables consumption compared to their counterparts, this might be due to the possibility of frequent contact with health professionals, which could expose them to sufficient nutritional counseling about feeding their children. This is in agreement with studies that report positive association of frequent ANC visits with dietary diversity of children age 6–23 months [37]. It is undoubtedly that ANC present an opportunity for enhancing maternal and child nutrition in low and middle income countries as during ANC education on healthy eating including the importance of fruits and vegetables in the diet is conducted, which may result in increased awareness of mothers about the importance of healthy diet for their own health and the health of their children [38]. This awareness may lead them to prioritize the feeding of fruits and vegetables that will lead to a reduction of zero fruits/vegetables consumption and eventually reduction in micronutrients deficiencies among children. Therefore, promoting frequent ANC visits could enhance dietary diversification, which could lead to increase fruits and vegetable consumption.

However, children who were born in health facility had higher probability of zero fruits/vegetables consumption as compared with their counterparts. This could be due to the fact that mothers with complicated pregnancies are more likely to deliver in a facility and complication during delivery is also linked with less frequent antenatal visits and mothers with less frequent ANC visits are less exposed to nutrition education.

Children aged 12–18 month, and 19–23 months had less probability of consuming zero fruits/vegetables compared to children aged 6–11 months. This is consistent with a study conducted in sub-Saharan Africa that revealed that children aged 6–11 months were less likely to

have consumptions of iron-rich foods including iron rich fruit and vegetables [39]. This observation is normal as children 6–11 are in a transition between liquid/semi liquid foods and the mainly solid family foods as they have limited chewing abilities. It could also be due to inappropriate complementary feeding practice among mothers of children aged 6–11 months old [35] or mother's perceptions that children before the age of 1 year should not consume fruits/vegetables.

This study also revealed that children of mothers who were Muslim were less likely to have zero fruits/vegetables consumption compared to children of mothers who practice orthodox religion. Although the feeding habit of a mother is influenced by many factors including culture, personal beliefs as well as, access to food, extensive fasting by mothers who are Orthodox Christians may influence complementary feeding practices [40]. We recommend that future research should qualitatively explore the feeding experience of women while fasting.

Furthermore, region which is a community level variable was also significantly associated with zero fruits/vegetables consumption. Children in the small peripheral regions (Afar, Benishangul, Gambella, and Somalia) were more likely to have zero fruits/vegetables consumption as compared with children from the metropolitan regions. This difference could due to easy access to fruits/vegetables, and better awareness of mothers on the benefits of feeding fruits/vegetables to children in Metropolitan areas. Furthermore, the discrepancy might be due to poor socioeconomic status of small peripheral regions that results from repeated drought [41].

This study has strength of addressing unaddressed topic in Ethiopia using nationally representative data as well as advanced modeling to estimate both individual, household and community level variables. However, due to the secondary nature of the data we are unable to incorporate more variables like accessibility to fruits and/vegetables, parental feeding habit and psychological variables. Despite this limitation we believe that the study could serve as a baseline for future researchers.

## Conclusion

In Ethiopia, the proportion of zero fruits/vegetables consumption among children age 6–23 months is high. Zero fruits/vegetables consumption was significantly associated with household wealth index, mother's age, mother's marital status, religion, child age, attending 4+ANC, place of delivery and region. Therefore, efforts should be made by stakeholders who are concerned about optimal diet and health of children to improve fruits and vegetables consumption of children particularly those from poor households, young mothers (15–24 years), and those from peripheral region of Ethiopia. This could be done during ANC follow up visits and during nutrition counseling.

## Supporting information

**S1 Checklist. STROBE statement—Checklist of items that should be included in reports of observational studies.**
(DOCX)

## Acknowledgments

We would like to extend our acknowledgment to the measure DHS for providing the data.

## Author Contributions

**Conceptualization:** Birhan Ewunu Semagn.

**Data curation:** Birhan Ewunu Semagn.

**Formal analysis:** Birhan Ewunu Semagn.

**Investigation:** Birhan Ewunu Semagn.

**Methodology:** Birhan Ewunu Semagn.

**Project administration:** Birhan Ewunu Semagn.

**Resources:** Birhan Ewunu Semagn.

**Software:** Birhan Ewunu Semagn.

**Supervision:** Abdulai Abubakari.

**Validation:** Birhan Ewunu Semagn, Abdulai Abubakari.

**Visualization:** Birhan Ewunu Semagn, Abdulai Abubakari.

**Writing – original draft:** Birhan Ewunu Semagn.

**Writing – review & editing:** Birhan Ewunu Semagn, Abdulai Abubakari.

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
