## [Decision Letter · Decision Letter 0]

27 Mar 2023

PONE-D-23-02355Magnitude of Zero Vegetable / Fruit Consumption and Associated Factors among Children age 6-23 months in Ethiopia: Mixed effect logistic regression analysisPLOS ONE

Dear Dr. Semagn,

Thank you for submitting your manuscript to PLOS ONE. After careful consideration, we feel that it has merit but does not fully meet PLOS ONE’s publication criteria as it currently stands. Therefore, we invite you to submit a revised version of the manuscript that addresses the points raised during the review process.

We look forward to receiving your revised manuscript.

Kind regards,

Melaku Kindie Yenit

Academic Editor

PLOS ONE

Journal Requirements:

Additional Editor Comments:

Dear Authors,

Thank you again for your manuscript submission.

Your manuscript has now been reviewed by experts in the field. Please have a look in the attachment section and act accordingly.

Reviewers' comments:

Reviewer's Responses to Questions

**Comments to the Author**

1. Is the manuscript technically sound, and do the data support the conclusions?

Reviewer #1: Partly

Reviewer #2: Yes

2. Has the statistical analysis been performed appropriately and rigorously? 

Reviewer #1: Yes

Reviewer #2: Yes

3. Have the authors made all data underlying the findings in their manuscript fully available?

Reviewer #1: Yes

Reviewer #2: No

4. Is the manuscript presented in an intelligible fashion and written in standard English?

Reviewer #1: No

Reviewer #2: No

5. Review Comments to the Author

Reviewer #1: Dear Authors,

Greetings!

I hereby testify that I am very much delighted to read and contribute my part for the betterment of your manuscript. The issue you raised is a new indicator for evaluating the under-two children's minimum dietary diversity status in the 2021 WHO/UNICEF IYCF Guideline. I found your work is scientifically relevant and will yield a baseline evidence for Ethiopia's IYCF status pertaining to vegetables/fruits consumption in under-two years children. Please be advised that you need to incorporate or take in to consideration of my issues or concerns while you work on the manuscript revision. Thanks!

1) Title: Have you assessed vegetable only or fruits only or both? What if some eats vegetable but not fruits and vice versa?

2) Please correct your Affiliation as: Amhara, not Ahara

3) How do you generate keywords? The keywords you stated are already presented in the title; what is the need to repeat them?

4) Please write as 'Methods' or 'Results'; not 'Method' or 'Result' in all sections of the manuscript

5) Where are the multi-level findings in the abstract section of the results?

6) Conclusion for the first objective is confusing; not focused

7) Background: 1) Not focused to the problem under study; 2) Existing gaps weren't well stated; 3) There is no information about existing factors so far

8) Methods: 1) You stated 9 regions, but Ethiopia has 11 regions, currently; you need to specify it; 2) P-value < or = 0.25 in the bivariable analysis were entered in to the multivariable analysis: why you do this since you have small number of variables?

9) Results: 1) What is your focus: 'Prevalence' or 'Magnitude' or 'Proportion'? 2) You have tables without heading: correct it; 3) There is discrepancies of results in the abstract and main body

10) Discussion: 1) Is that, 'zero fruit and vegetables' or 'zero fruits or vegetables'? Be consistent in your wording; 2) Very shallow discussion [it has no implications to nutrition interventions, clinical settings, research, policy, and programs]: please include these in the discussion

11) Conclusion: 1) There is discrepancies in the main body and abstract section of the manuscript [especially, your first objective is somewhat different in both sections]: focus on your objectives, and findings while giving concluding remarks

12) Ethical considerations: 1) Ethical approval is required for research projects or manuscripts that used secondary data that includes personal data [data that relates to identifiable living persons]. However, your statement says, ethical approval was not your issue. How do you comment on this?

13) Abbreviations: The expanded form for 'ANC' is Antenatal Care, not Ante Natal Care: Please correct that one

14) General comments: 1) The manuscript is highly affected by grammatical and punctuation errors. Therefore, it is highly recommended to consult an English Expert who will work on standard and plain English so that you manuscript will be more suitable for publication and reading; 2) There is inconsistency of using major keywords throughout the document; please be serious on this issue, and try to use same wording all the way through; 3) There is slight difference in the affiliation statements between the online submission and in the manuscript; try to make them identical

Reviewer #2: Summary: dig out such developing countries problem is important but showing the magnitude at local level is more significant, because the data obtained from this research finding is almost defined in the DHS report also.

Background: line 63: better to rephrase it.

Methodology: Line 146-149: you mentioned a contradictory idea with your justification. Amendment on your justification is required. If it is so your study doesn’t show any new findings than the previous studies you cited. Do you think wealth index is an individual level variable? Did you generate WI again or you took from the report of EMDHS? When you classify poor, middle or rich, the minimum required number??? When we say there is media exposure, better to operationalized???

Line 161: what is your weighting number???

Line165: Have checked it and violated the assumptions or based on your statistical knowledge?

Line 172-73: the ultimate use of LLR, deviance or AIC is similar. Why not you used either of one???

Line 193: large central region??? Better to operationalize.

Result, discussion

Line 198: Most of sample were from poor house household….. Will biased your findings. Needs management. Narration and table similar…..

Line 206: Is the 95% confidence interval wide or narrow? Why?

Line 215-table: 1-have checked chi-square assumptions? And if the difference in the geopolitical regions were significant, what did you think about the regression you did?

Line 228-229: Does the interpretation give a logical sense???

Table 2: if you obtain similar values for them, and between them, which is better for multilevel model comparison if any?

Table 3: Your statistical significant variables are mostly sociodemographic and economic factors. What do you feel about these and its implications?

Line 281-285: Dear authors, are you sure your findings are consistent with the cited reference findings??? If it is so, what is the criteria to say consistent, inline, higher, lower, opposite etc.?

Line 289-291: if the participants were different, can we use them for comparison?

Line 296: most of your participants are from poor households…… it is difficult to assure a true statistical significance……… if the sample size for all categories is high, it might not be significant. Do you have any method you used for statistical significant assurance???

Line 329: Your interpretation is inappropriate, the religion you mentioned does not recommend fasting for populations like your participants. Needs revision or better justification.

6. PLOS authors have the option to publish the peer review history of their article (what does this mean?). If published, this will include your full peer review and any attached files.

Reviewer #1: No

Reviewer #2: No

---

## [Author Response · Author response to Decision Letter 0]

26 Apr 2023

Academic Editor:Thank you for giving us the opportunity to submit a revised draft of the manuscript . All the concerns raised by the experts are addressed point by point.

Reviewer 1:We have incorporated all of your suggestions in to the revised manuscript. They were very helpful. Thank you!

Reviewer 2:We have incorporated all of your suggestions in to the revised manuscript. They were very helpful. Thank you!

---

## [Decision Letter · Decision Letter 1]

4 Jul 2023

Zero fruits/vegetables consumption and associated factors among Children aged 6-23 months in Ethiopia: Mixed effect logistic regression analysis

PONE-D-23-02355R1

Dear Dr. Semagn,

We’re pleased to inform you that your manuscript has been judged scientifically suitable for publication and will be formally accepted for publication once it meets all outstanding technical requirements.

Kind regards,

Miquel Vall-llosera Camps

Senior Editor

PLOS ONE

Reviewers' comments:

Reviewer's Responses to Questions

**Comments to the Author**

1. If the authors have adequately addressed your comments raised in a previous round of review and you feel that this manuscript is now acceptable for publication, you may indicate that here to bypass the “Comments to the Author” section, enter your conflict of interest statement in the “Confidential to Editor” section, and submit your "Accept" recommendation.

Reviewer #1: All comments have been addressed

Reviewer #2: All comments have been addressed

2. Is the manuscript technically sound, and do the data support the conclusions?

Reviewer #1: Yes

Reviewer #2: Yes

3. Has the statistical analysis been performed appropriately and rigorously? 

Reviewer #1: Yes

Reviewer #2: Yes

4. Have the authors made all data underlying the findings in their manuscript fully available?

Reviewer #1: Yes

Reviewer #2: Yes

5. Is the manuscript presented in an intelligible fashion and written in standard English?

Reviewer #1: Yes

Reviewer #2: No

6. Review Comments to the Author

Reviewer #1: I have seen that all comments were corrected. Therefore, I would recommend the editors and/or PLOS ONE could publish this article.

Reviewer #2: (No Response)

7. PLOS authors have the option to publish the peer review history of their article (what does this mean?). If published, this will include your full peer review and any attached files.

Reviewer #1: No

Reviewer #2: No

---

## [Editor Report · Acceptance letter]

7 Jul 2023

PONE-D-23-02355R1 

Zero fruits/vegetables consumption and associated factors among Children aged 6-23 months in Ethiopia: Mixed effect logistic regression analysis 

Dear Dr. Semagn:

I'm pleased to inform you that your manuscript has been deemed suitable for publication in PLOS ONE. Congratulations! Your manuscript is now with our production department. 

Kind regards, 

on behalf of

Dr. Miquel Vall-llosera Camps 

Staff Editor

PLOS ONE